# Dietary and Pharmacological Treatment of Nonalcoholic Fatty Liver Disease

**DOI:** 10.3390/medicina55050166

**Published:** 2019-05-20

**Authors:** Anna Jeznach-Steinhagen, Joanna Ostrowska, Aneta Czerwonogrodzka-Senczyna, Iwona Boniecka, Urszula Shahnazaryan, Alina Kuryłowicz

**Affiliations:** 1Clinical Dietetics Department, Medical University of Warsaw, 01-445 Warsaw, Poland; joanna.maria.ostrowska@gmail.com (J.O.); aneta.senczyna@gmail.com (A.C.-S.); iboniecka@gmail.com (I.B.); 2Diabetologic Outpatients Department, Institute of Mother and Child, 01-211 Warsaw, Poland; 3Department of General, Oncological, Metabolic and Thoracic Surgery, Military Institute of Aviation Medicine, 01-755 Warsaw, Poland; 4Department of Internal Diseases and Endocrinology, Medical University of Warsaw, 02-097 Warsaw, Poland; ulashahna@gmail.com (U.S.); akurylowicz@imdik.pan.pl (A.K.)

**Keywords:** nonalcoholic fatty liver disease (NAFLD), antidiabetic agents, diet, physical activity

## Abstract

Nonalcoholic fatty liver disease (NAFLD) is the most common chronic liver disease in the developed world. Simple hepatic steatosis is mild, but the coexistence of steatohepatitis (NASH) and fibrosis increases the risk of hepatocellular carcinoma. Proper dietary and pharmacological treatment is essential for preventing NAFLD progression. The first-line treatment should include dietary intervention and increased physical activity. The diet should be based on the food pyramid, with a choice of products with low glycemic index, complex carbohydrates in the form of low-processed cereal products, vegetables, and protein-rich products. Usage of insulin-sensitizing substances, pro- and prebiotics, and vitamins should also be considered. Such a therapeutic process is intended to support both liver disease and obesity-related pathologies, including insulin resistance, diabetes, dyslipidemia, and blood hypertension. In the pharmacological treatment of NAFLD, apart from pioglitazone, there are new classes of antidiabetic drugs that are of value, such as glucagon-like peptide 1 analogs and sodium/glucose cotransporter 2 antagonists, while several other compounds that target different pathogenic pathways are currently being tested in clinical trials. Liver biopsies should only be considered when there is a lack of decline in liver enzymes after 6 months of the abovementioned treatment. Dietary intervention is recommended in all patients with NAFLD, while pharmacological treatment is recommended especially for those with NASH and showing significant fibrosis in a biopsy.

## 1. Introduction

Nonalcoholic fatty liver disease (NAFLD) is a chronic liver disease, diagnosed if secondary causes of steatosis—such as excessive alcohol consumption, long-term use of steatogenic substances, and monogenic liver disorders—can be excluded. NAFLD is characterized by morphological and biochemical changes typical for alcoholic steatohepatitis, but present in individuals consuming less than 20 g of alcohol per day for women, and less than 30 g of alcohol per day for man. NAFLD refers to the nonalcoholic fatty liver (NAFL) that can be accompanied by mild lobular hepatitis, as well as to nonalcoholic steatohepatitis (NASH). Subsequently, NASH includes early NASH without or with insignificant fibrosis (grades F0–F1), scirrhous NASH with significant fibrosis (F ≥ 2) or advanced fibrosis (F ≥ 3), NASH with cirrhosis (F4), and hepatocellular carcinoma (HCC). Diagnosis of all forms relies on excessive (over 5% of the dry organ mass) accumulation of triglycerides in the liver. For NASH, an inflammatory process that can cause lobular hepatitis, hepatocellular ballooning, and fibrosis are typical [1]. 

NAFLD is currently the most prevalent chronic liver disease in high-growth countries. It is diagnosed in 15%–40% of the adult population (with differences according to the diagnostic method, age, sex ethnicity, and body mass index (BMI), with a frequency twice more often in men than in women.) [2]. The main reason for underdiagnosis is that the disease is asymptomatic or accompanied by non-specific symptoms. Diagnosis of NAFLD should be considered in every patient with mild aspartate aminotransferase (AST) and/or alanine aminotransferase (ALT) elevation (rarely above 300 IU/l) after excluding other causes of liver disease. In most cases, ALT levels are higher than AST. Given the risk of progression, non-invasive methods of fibrosis evaluation are of great importance. Ultrasonography, combined with elastography, was found to be a useful tool for diagnosis of NAFLD, with 87% sensitivity and 91% specificity [3]. Liver biopsy and histopathological evaluation are performed to differentiate simple NAFLD from NASH or fibrosis, but also to predict the disease course. 

Multiple factors might modify the NAFLD course. About 90% of NAFLD patients present at least one feature of the metabolic syndrome, while the fully symptomatic metabolic syndrome (diagnosed in 33% patients) is a significant risk factor for the progression from NAFL into NASH [4]. In addition, co-existence of other medical conditions, such as hypothyroidism, hypogonadism, obstructive sleep apnea, polycystic ovary syndrome, and celiac disease, may associate with or promote the progress of NAFLD into NASH [5,6,7,8,9]. Intestinal barrier damage and intestinal microbiota imbalance (lower concentration of Bacteroidetes and higher concentration of Firmicutes) may also promote the progression of the disease [10]. According to the “multiple hit” hypothesis, in obese individuals with excessive lipid accumulation in the liver (“first hit”), derangement of the gut–liver axis acts as a “second hit” leading to the progression of liver damage, inflammation, and subsequent fibrosis [11]. Intestinal dysbiosis might contribute to NAFLD progression in several ways: (i) by increasing energy extraction from diet due to the altered capacity to digest and ferment complex polysaccharides; (ii) by damage caused to intestinal epithelium by bacterial production of ethanol; (iii) by translocation of bacteria and endotoxins to the portal circulation and activation of pro-inflammatory signaling via toll-like receptors (TLRs); (iv) by modifications of bile acid synthesis; (v) by a decrease in choline metabolism, resulting in reduced liver export of very low density lipoproteins [12].

Given its high frequency and related health consequences, NAFLD should be treated upon the moment of diagnosis. In this narrative review, based on the current guidelines supplemented with a review of the latest literature, we present possible therapeutic options that include lifestyle interventions and pharmacological treatment.

## 2. Lifestyle Interventions

Non-pharmacological treatment remains a first-line strategy in NAFLD management. Therefore, it is recommended for all patients that they change their eating habits. The goals of treatment include weight reduction, prevention of metabolic syndrome, and NAFLD progression. Given the high prevalence of overweight/obesity and diabetes in NAFLD patients, attention should be paid to the glycemic index (IG) and the energy value of products. Favorable effects were also observed in patients on a Mediterranean diet and a diet based on the healthy food pyramid, including low and medium IG products [13]. Liver biopsy should only be considered in the case a lack of decline of liver enzymes during the 6 months of first-line treatment.

Based on the available literature, we summarized dietary recommendations for patients with NAFLD in Table 1.

### 2.1. Caloricity of Diet and Weight-Reduction Goals

In overweight (body mass index, BMI = 25–29.9 kg/m^2^) and obese (BMI ≥ 30 kg/m^2^) patients weight reduction by 5%–10% results in a 20%–80% decrease in intrahepatic triglyceride content and serum aminotransferase activity [1,7,8,34,35]. The optimal weight loss goal seems to be 0.5–1.0 kg per week (although some authors suggest higher goals), with particular attention to visceral fat reduction [9,36,37]. In patients with NAFLD, reduction of 5% to 7% of the initial body weight is recommended, while in those with suspected or biopsy-proven NASH, the weight loss goal is higher (7% to 10%) [14]. If the target weight loss does not lead to the required decrease in aminotransferase activity, additional weight loss is suggested. It is worth mentioning that very low-calorie diets (VLCD), providing about 500 kcal per day, are not recommended in NAFLD management since, despite reduction in hepatic steatosis, they led to the exacerbation of fibrosis and necrosis caused by excessive weight loss [10,11]. In obese patients who do not meet weight loss goals after 6 months of lifestyle intervention, bariatric surgery may be considered [1,38]. This treatment is recommended in patients with NASH or advanced fibrosis, but not with decompensated cirrhosis. Surgically induced weight loss leads to histological improvement but may sometimes result in worsening of fibrosis; therefore, postoperative monitoring is required [12].

### 2.2. Dietary Fats

Reduction of saturated fats consumption to less than 7% is beneficial for patients with normal glucose tolerance but, however, does not improve the lipid profile and may even be detrimental to individuals with insulin resistance [19,20]. A diet rich in saturated fats leads to intensified oxidative stress and, consequently, to the development of inflammation in the area of simple hepatic steatosis and hepatocytic damage. High consumption of *trans* fat (bakery fat, fast food margarine, instant meals, cakes) increases the cardiovascular risk but also enlarges the hepatic mass as a result of excessive cholesterol and triglyceride accumulation [39]. In turn, polyunsaturated fatty acids (PUFA) from the omega-3 group (n-3) were found to improve serum lipid parameters, and decrease hepatic steatosis and transaminase activity after just one-year observation [40,41]. However, in randomized trials, only the decrease of hepatic steatosis was confirmed [42]. 

Beneficial effects were also reported after supplementation with monounsaturated fatty acids (MFA) that, via stimulation of peroxisome proliferator-activated receptors (PPARα and PPARγ), increase lipid oxidation and contribute to the decrease in the accumulation of triglycerides in the liver [43].

### 2.3. Dietary Carbohydrates

A decrease in the consumption of simple carbohydrates and complete exclusion of added sugar play a key role in the treatment and prevention of NAFLD. There is also growing evidence that an excess supply of fructose and sucrose (especially from sweet beverages) promotes the development of metabolic disorders [44]. Diets based on high IG products intensify liver steatosis, particularly in patients with existing insulin resistance, by increasing lipogenesis and triglyceride deposition in hepatocytes [17,18]. 

### 2.4. Dietary Protein

One of the causes of NASH may also be a low protein diet. Protein is essential for rebuilding damaged hepatocytes and providing the methionine and choline necessary for incorporation of lipids into lipoproteins that prevent fat accumulation in the liver. Rich protein diets (providing approximately 40% of energy from protein) combined with physical activity are more effective in reducing fat content and body weight, as well as in improving lipid profile, than low protein diets (providing 15% of energy from protein) and diets based on carbohydrates (providing 55% of energy from sugars), however, higher protein intake may have adverse effects on kidney function and bone turnover [21,22]. 

### 2.5. Antioxidants

Anthocyanins (extracted from blackberries and blackcurrants) and resveratrol have beneficial effects in NAFLD patients thanks to their antioxidant potential, as observed by a decrease in triglyceride levels, suppression of hepatic steatosis and hepatocyte apoptosis, and reduction of hepatic inflammation and insulin resistance [27]. Cinnamon and turmeric are also believed to improve insulin sensitivity, decrease fasting glucose levels by reduction of hepatic gluconeogenesis, lower Homeostatic Model Assessment–Insulin Resistance (HOMA-IR), improve lipid profile and decrease transaminase activity [28,29,45].

### 2.6. Probiotics and Prebiotics

In interventional studies, supplementation with probiotics (*Lactobacillus* and *Bifidobacterium*) efficiently reduced fatty acid synthesis, metabolic endotoxemia and inflammation in animal models of NAFLD [10]. In meta-analyses, probiotic therapies effectively reduced aminotransferases, total cholesterol, triglycerides, and pro-inflammatory cytokine serum levels as well as improved insulin sensitivity and ultrasound liver image in NAFLD patients [11,46,47]. However, the effect of the therapy may vary depending on the bacterial strains and regimen of treatment [47]. Prebiotics have also been found to have a direct influence on lipid and carbohydrate metabolism. Beneficial effects of, e.g., oligofructose and inulin on glucose, glycated hemoglobin, triglycerides, and total and low-density lipoprotein (LDL) cholesterol level, as well as on transaminase activity, in patients with type 2 diabetes and NAFLD were observed [10,31]. Additionally, oligofructose supplementation promoted weight loss irrespective of patients’ lifestyle [32]. Similarly, in a randomized trial, the addition of synbiotics (combining probiotics and prebiotics) in the lifestyle intervention led to a significant reduction in hepatic steatosis and fibrosis, and decrease in serum glucose, triglycerides, and inflammatory mediator levels [48].

### 2.7. Alcohol Consumption

An essential aspect of NAFLD management is the reduction of alcohol consumption. Alcohol abuse (>14 drinks per week) is positively associated with disease progression, while data regarding the influence of moderate alcohol consumption on NAFLD course are univocal. In this situation, abstinence from alcohol is recommended [33].

### 2.8. Physical Activity

Exercise supports the process of weight loss and allows maintenance of physical capacity. All NAFLD patients should perform aerobic exercise for at least 150 min a week, preferably 30 min a day. Moderate intensity aerobic training (walking, cycling) and resistance training are recommended to improve mobility and metabolic parameters. High-intensity interval training was also effective in weight reduction and improvement of metabolic parameters, including lipid profile, alanine aminotransferase activity, and hepatic fat content [49,50]. After attaining the appropriate body weight, it is advised to continue exercise in order to preserve the obtained results. 

## 3. Pharmacological Treatment

The lack of specific pharmacological recommendations with proven efficacy makes NAFLD management a complicated process. Therefore, treatment is focused on associated/co-existing diseases (diabetes, obesity, lipid disorders) to control the patient’s glycaemia, liver function, and lipid profile. Pharmacological therapy is recommended for individuals who do not achieve expected weight loss goals, and for those with NASH with fibrosis stage ≥ 2 (F2) in biopsy [51].

### 3.1. Vitamin Supplementation

Vitamin supplementation was found to be essential for NAFLD management. Vitamins with antioxidant properties, such as vitamin E and C, were found to decrease ALT and AST serum activity, as well as decrease lipoatrophy and lobular hepatitis without affecting liver fibrosis [23]; therefore, their supplementation should be considered as a first choice therapy in patients with NASH and fibrosis stage ≥ 2 (F2) as proven by biopsy [1,2,23,24]. In non-diabetic patients, a daily dosage of 800 IU of vitamin E is recommended. Since diabetic patients were not included into the studies assessing the efficacy of vitamin E treatment in NASH, the American Association for the Study of Liver Diseases (AASLD) does not recommend such treatment in individuals with co-existing diabetes [1]. It should be mentioned that in some studies, administration of high (≥400 IU per day) doses of vitamin E was associated with an increase in all-cause mortality as well as with the progression of prostate cancer and, therefore, it is not recommended in individuals without NASH and F2 and in those with personal history or strong family history of prostate cancer [25]. Adequate intake of vitamin D also plays a role in the prevention of NAFLD, and has proven efficacy in increasing insulin sensitivity and in lowering aminotransferase activity [26]. 

Among other substances with antioxidant properties, glutathione was found to decrease ALT levels and hepatic steatosis in NAFLD patients; however, large-scale clinical trials are needed to verify this finding [52].

### 3.2. Antidiabetic Drugs

Several antidiabetic drugs were found to be effective in NAFLD/NASH management. A significant reduction of hepatic steatosis was reported after treatment with pioglitazone (12–72 weeks), liraglutide, a glucagon-like peptide 1 (GLP-1) agonist (26–50 weeks), and insulin combined with metformin (3–7 months) [53,54,55]. Additionally, in placebo-controlled trials, pioglitazone, liraglutide, and exenatide were effective in the reduction of liver fibrosis [53,54,56]. Treatment with metformin or dapagliflozin (sodium/glucose cotransporter 2 antagonist, SGLT-2) alone did not have a positive effect on liver histology [57], while other SGLT-2 inhibitors were found to suppresses hepatic accumulation of triglycerides (ipragliflozin) [58] and to reduce inflammatory marker levels and aminotransferases activity (empagliflozin) [59]. The therapeutic options available for patients with type 2 diabetes are summarized in Table 2. 

### 3.3. Compounds Modifying Lipid Profiles

To limit the risk of cardiovascular events, NAFLD patients with dyslipidemia should be treated with statins or ezetimibe; however, these drugs have no significant effect on liver histopathology [62].

Several novel compounds interfering with lipid metabolism are being tested for their efficacy in NAFLD treatment. Aramchol, a cholic–arachidic acid conjugate, acting as a stearoyl-CoA desaturase (SCD) inhibitor was found to reduce hepatic fat accumulation both in animal and in human studies [63], and the drug is being tested on biopsy-proven NASH patients without cirrhosis in a phase 2b clinical trial [64]. GS0976, an inhibitor of the acetyl-CoA carboxylase (ACC), a key regulator of fatty acid metabolism, was also found to decrease hepatic fat content and levels of serum markers of hepatic fibrosis in NASH patients [64].

Interference with liver lipid metabolism, in order to improve the course of NAFLD, has been extensively investigated in preclinical studies, e.g., through inhibition of diacylglycerol acyltransferase 2 (DGAT) that catalyzes the final step in triglyceride synthesis, and with antisense oligonucleotides in rats with diet-induced NAFLD, resulting in significantly reduced hepatic steatosis and improved insulin sensitivity [65]. This concept may represent a promising therapeutic perspective.

### 3.4. Anti-Obesity Compounds

Until now, there has not been any reports demonstrating that medications registered for obesity treatment, such as naltrexone/bupropion, phentermine/topiramate, or lorcaserin, have any effect on NASH, although their impact on weight loss may indirectly influence NAFLD course. A randomized, double-blind study has also demonstrated the beneficial effect of orlistat on ALT activity and the reduction of liver fibrosis [66]. 

### 3.5. Novel Therapeutic Perspectives

In recent years, novel mechanisms in the pathogenesis of NAFLD have been revealed, and created novel therapeutic possibilities for NAFLD treatment that include, among others, activation of farnesoid X receptor (FXR) as well as interference with apoptotic, fibrotic, and inflammatory pathways.

### 3.6. Farnesoid X Receptor Agonists

FXR is a nuclear receptor that is widely expressed in liver and regulated by bile acids, whose abnormal activity is associated with NAFLD. Obeticholic acid (OCA) is a first-in-class selective FXR agonist with anticholestatic and hepatoprotective properties, and registered for the treatment of primary biliary cholangitis [67]. In phase 3 clinical trials performed in patients with NAFLD and type 2 diabetes and in individuals with NASH proven by biopsy, OCA was found to enhance insulin sensitivity, control glucose homeostasis, modulate lipid metabolism, and exert anti-inflammatory and antifibrotic effects in the liver. The most common adverse effects of OCA treatment include gastrointestinal problems, pruritus, fatigue, headache, increase in LDL cholesterol, and a decrease in high-density lipoprotein (HDL) cholesterol and triglycerides. [67]. Therefore, synthetic non-bile acid FXR agonists, that have the potential to provide favorable metabolic effects without increasing these side effects, are currently being assessed in phase 2 trials [64].

Another bile acid, ursodeoxycholic acid (UDCA), produced naturally by intestine bacteria, has also been tested for its utility in NAFLD treatment. In preclinical studies, UDCA was found to exert anti-apoptotic, antioxidant, and anti-inflammatory effects. However, in clinical trials in patients with NASH, UDCA has failed to have a significant influence on liver inflammation or fibrosis and, therefore, it is not currently recommended by the guidelines [1,68]. Nevertheless, it has been suggested that a combination of UDCA with other agents, such as vitamin E and/or omega-3, might have an additive effect in diminishing NASH-associated fibrosis [69].

### 3.7. Dual PPAR Agonists

Another new trend in NAFLD treatment is represented by dual PPAR agonists. Saroglitazar, a dual PPARα/γ agonist used for the treatment of dyslipidemia in diabetic patients, and elafibranor, a dual agonist of PPARα/δ which was found to be effective in improvement of steatosis, inflammation, and fibrosis in mouse models of NAFLD, are currently under assessment in phase 2 trials [64]. 

### 3.8. Compounds Interfering with Apoptotic Pathways

Anti-apoptotic agents such as selonsertib, an inhibitor of the apoptosis signal-regulating kinase 1 (ASK1—a key player in the pathways leading to liver apoptosis and fibrosis) was evaluated in phase 2 trials in NASH patients with moderate-to-severe liver fibrosis (stages 2/3). Treatment led to a significant regression of fibrosis stage, reduction of liver stiffness and fat content, and diminished the risk of progression to cirrhosis. Thus, international phase 3 trials evaluating selonsertib in NASH patients with stage 3 are ongoing [64].

In turn, mTORC1 inhibitor (rapamycin) was found to improve liver steatosis in high-fat diet-fed mice; however, treatment resulted in increased production of interleukin 6 and activation of the signal transducer and activator of transcription 3 (STAT3) pathway that may enhance HCC development [70].

### 3.9. Antifibrotic and Anti-Inflammatory Compounds

Since fibrosis stage determines mortality in NASH patients, effective antifibrotic treatment could improve the course and prognosis of the disease. Therefore, several compounds with anti-inflammatory and antifibrotic potential have been considered for the treatment of advanced NASH.

Historically, pentoxifylline (a non-selective phosphodiesterase inhibitor) that exerts an anti-inflammatory effect via inhibition of tumor necrosis alpha (TNF-α) and, in this way, may decrease aminotransferase activity, is controversial, as well as the way this drug influences the histological image of the liver [71,72].

Nowadays, the antifibrotic compound emricasan, a caspase inhibitor that improves fibrosis in murine models of NASH, is being evaluated for its utility in a phase 2b study in patients with NASH (stage 1–3), including those with cirrhosis and severe portal hypertension [73]. A similar antifibrotic effect was reported in murine models for the vascular adhesion protein-1 (VAP-1) inhibitors. VAP-1 is responsible for transmission of profibrogenic and pro-inflammatory stimuli. Thus, inhibitors targeting VAP-1 are under evaluation in clinical trials in NASH patients [74].

Another anti-inflammatory and antifibrotic compound, cenicriviroc (CVC), a C–C motif chemokine receptor-2/5 (CCR2/5) antagonist that also improves insulin sensitivity via inhibition of macrophage recruitment into adipose tissue, was reported to improve fibrosis without worsening NASH when compared with placebo [75], and is being further tested for clinical application.

Available and future therapies for different stages of NAFLD are summarized in Figure 1.

## 4. Final Remarks and Conclusions

In developed countries, NAFLD/NASH is the most common cause of liver cirrhosis and HCC due to the growing epidemic of obesity and metabolic syndrome. Due to its long-term consequences, it should be immediately treated upon diagnosis. The therapeutic process should be aimed at both liver disease and obesity-related pathologies (insulin resistance, diabetes, and dyslipidemia). The first-line treatment should include lifestyle interventions regarding diet (based on the food pyramid with a choice of products with low and medium IG, complex carbohydrates in the form of low-processed cereal products, vegetables, and protein-rich products) and increased physical activity. 

Liver biopsy should be performed if 6 months of the first-line treatment does not lead to a decline of liver enzymes. In patients with NASH with fibrosis stage ≥ 2 (F2) in biopsy, supplementation of high doses of vitamin E and treatment with antidiabetic drugs, such as pioglitazone, GLP-1 analogs, and SGLT-2 antagonists, should be considered. Knowledge of novel mechanisms in the pathogenesis of NAFLD created novel therapeutic possibilities for its treatment; therefore, several compounds are being tested in trials for their utility in clinical practice.

## Figures and Tables

**Figure 1 medicina-55-00166-f001:**
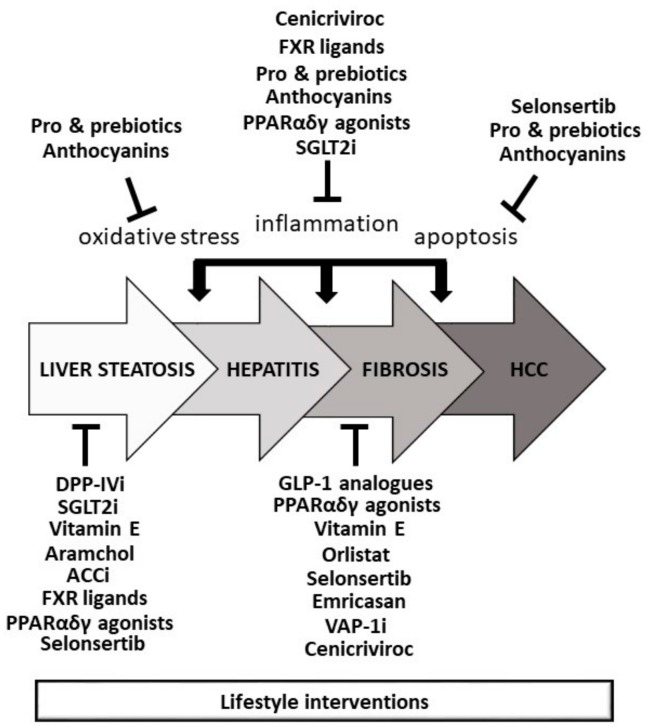
Available and future therapies for different stages of nonalcoholic fatty liver disease (modified based on Sumida & Yoneda, 2018 [64]). ACCi—acetyl-CoA carboxylase inhibitors, DPP-IVi—dipeptidyl peptidase-4 inhibitors, FXR—farnesoid X receptor, GLP-1—glucagon-like peptide-1, HCC—hepatocellular carcinoma, PPARs—peroxisome proliferator-activated receptors, SGLT2i—sodium glucose co-transporter 2 inhibitors, VAP-1i—vascular adhesion protein-1 inhibitors.

**Table medicina-55-00166-t001a:** A. Weight-reduction goals and caloricity of the diet.

Parameter	Value/Goal
Expected reduction of the initial body weight within 6 months [14]	In patients with: - NAFL—5% to 7% - suspected or biopsy-proven NASH—7% to 10%
The expected rate of weight loss [1,15]	- optimal 0.5–1 kg / week - in severe obesity even > 1.5 kg/week
Caloricity of the diet [16]	- women: 1200–1500 kcal per day - men: 1500–1800 kcal per day or decrease the daily demand by 25% (i.e., by 500–1000 kcal) - diets providing < 500 kcal per day are not recommended

**Table medicina-55-00166-t001b:** B. Composition of the diet.

Parameter	Value	Comment
Reduction of carbohydrate intake [17,18]	40%–50% of energy	
Reduction of simple carbohydrates with a particular focus on fructose [17,18]	<10% of energy	**Simple carbohydrates:** sweets, pastries, honey, fruits, drinks and fruit juices, fruit preserves (jams, syrups), cheese and flavored yoghurts
Reduction of dietary fat [16]	30% of energy	
Reduction of saturated fatty acids [19,20]	<7% of energy	**Saturated fatty acids:** butter, palm oil, cream, cream cheese (Philadelphia type), lard, bacon, hard margarine, bacon, chocolate and chocolates, rennet cheese
Reduction of *trans* fatty acids [20]	<1g/day	***trans* Fatty acids:** baking fats, confectionery and fats used for repeatedly frying, confectionery products, cookies, bars and fast food products, hard margarine
Increase of in protein intake [21,22]	15%–20% of energy	**Protein:** poultry, beef, veal, cottage cheese, eggs, fish (tuna, trout, mackerel, salmon, anchovies)
Antioxidants [23,24,25,26,27,28,29]		**Vitamin C:** paprika, sauerkraut, strawberries, blackcurrants, parsley, grapefruit, mandarin, raspberry, spinach
		**Vitamin E:** oils, sunflower seeds, almonds, pumpkin seeds, hazelnuts, peanuts, soft margarine
		**Antioxidants:** citrus, berries, grapes, *Brassica*, grains, kernels, nuts, unprocessed cereal products, spices and seasonings
Probiotics and prebiotics [30,31,32]		**Probiotics and prebiotics:** fermented milk drinks (yogurts, kefir, buttermilk), garlic, chicory, artichokes, asparagus, onion
Reduction of alcohol intake [1,33]	<30 g/d for men <20 g/d for women	Abstinence from alcohol is recommended

**Table 2 medicina-55-00166-t002:** Mechanism of action, effects, and side effects of medicines used for nonalcoholic fatty liver disease (NAFLD) treatment in diabetic patients.

Medicine	Mechanism of Action	Influence On				Significant Side Effects	Comment
		Body Weight	ALT Activity	Liver Steatosis *	Liver Histology		
**Metformin**	via AMPK-dependent and independent pathways: - ↓ liver gluconeogenesis - ↑ peripheral glucose uptake - improves the cardiovascular system - protects against cancer	↓	↓	↓, ↔	↔	- gastrointestinal problems - lactate acidosis	- despite no specific influence on liver histology is recommended in NAFLD/NASH patients with type 2 diabetes due to its beneficial pleiotropic effects, with the exception of those with advanced cirrhosis [55,60]
**Pioglitazone** **(PPARs ligand)**	- alters transcription of key genes of carbohydrate and lipid metabolism - ↑ peripheral glucose uptake - ↓gluconeogenesis in the liver	↑	↓	↓	improvement	- swelling - a tendency to weight gain	- improves hepatic ballooning degeneration, lobular inflammation and fibrosis, and seems to modify the natural course of NASH [53]
**GLP-1 receptor agonists**	- ↑ postprandial insulin secretion - ↓ postprandial glucagon secretion - ↑ satiety	↓	↓	↓	improvement	- upper respiratory tract infections - headaches - gastrointestinal problems	- liraglutide led to significant NASH resolution; both liraglutide and exenatide stopped the progression of liver fibrosis [54,56]
**DPP-IV inhibitors**	- ↓ GLP-1 and GIP degradation	↔	↓	↓	no data	- upper respiratory tract infections - headaches - gastrointestinal problems	- sitagliptin and vildagliptin reduce aminotransferase activity, vildagliptin reduces liver steatosis as assessed by imaging studies [61]
**SGLT-2 inhibitors**	- ↓ renal glucose reabsorption	↓	↓	↔	improvement	- urogenital infections	- leads to weight reduction and can help patients with diabetes reach therapeutic goals - canagliflozin suppresses hepatic accumulation of triglycerides [58] - empagliflozin reduces inflammatory markers levels and aminotransferase activity [59]

* assessed in imaging studies (ultrasound, computed tomography, or magnetic resonance) ↔ no change; ↓ decrease; ↑ increase ALT—alanine aminotransferase; AMPK—AMP-activated protein kinase; NASH—nonalcoholic steatohepatitis; PPARs—peroxisome proliferator-activated receptors; GLP-1—glucagon-like peptide-1; DPP-IV—dipeptidyl peptidase-4; GIP—glucose-dependent insulinotropic peptide; SGLT-2—sodium glucose co-transporter 2.

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
