# Peer review of "Dietary and Pharmacological Treatment of Nonalcoholic Fatty Liver Disease"

_medicina, 2019, doi:10.3390/medicina55050166_

Reviewer 1 Report

To the Authors

The topic is interesting and the paper is well written, with clear language. However, some points need to be clarified. I would suggest to the authors the to revise some points before resubmitting to the journal.

Comment:

Even if the work is well done, I believe  it would benefit from minor revisions.

-Page 1, line 38, please specific “less than 20 g  per day for women and 30    

  for man

-Page 2, line 47, please check and specify “15-40% “ and specific “ with

  differences according to the diagnostic method, age, sex ethnicity”

-Page 2, line 50, it would be better to append “confirmed and after exclusion of other causes

  of liver disease”

-Page 1, line 62, please  insert the reference.

-Table 1insert the references

-I would also transfer the text from line 120 to 133 on pag. 4 where you can find the paragraph entitled “pharmacological  treatment”.      

-Please update references

 Author Response

Reviewer #1

Following the Reviewer suggestion we introduced several changes into the text:

1)      Page 1, line 38, please specific “less than 20 g per day for women and 30 for man”

“It is characterised by morphological and biochemical changes typical for alcoholic steatohepatitis but present in individuals consuming less than 20g of alcohol per day for women and less than 30g of alcohol per day for a man.” (Page 1, lines 38-40)

2)      Page 2, line 47, please check and specify "15-40% " and specific " with differences according to the diagnostic method, age, sex ethnicity."

“NAFLD is currently the most prevalent chronic liver disease in high-growth countries. It is diagnosed in 15-40%, of the adult population with differences according to the diagnostic method, age, sex ethnicity, twice more often in men than in women [2].” (Page 2, lines 48-50)

3)    Page 2, line 50, it would be better to append “after exclusion of other causes of liver disease."

“Diagnosis of NAFLD should be considered in every patient with mild aspartate aminotransferase (AST) or/and alanine aminotransferase (ALT) elevation (rarely above 300 IU/l) after exclusion of other causes of liver disease” (Page 2, lines 52-54)

4)      Page 2, line 62, please insert the reference.

“In addition, co-existence of other medical conditions such as hypothyroidism, hypogonadism, obstructive sleep apnea, polycystic ovary syndrome and celiac disease may promote the progress of NAFL into NASH [5-9].” (Page 2, lines 63-65)

Following references [5-9] were added to the references list:

5. Kim, D., Kim, W., Joo, S.K., Bae, J.M., Kim, J.H., Ahmed, A. Subclinical Hypothyroidism and Low-Normal Thyroid Function Are Associated With Nonalcoholic Steatohepatitis and Fibrosis. Clin Gastroenterol Hepatol 2018, 16, 123-131.e1. Available online: https://www.ncbi.nlm.nih.gov/pubmed/28823829 (accessed on 11.02.2019).

6. Marino, L., Jornayvaz, F.R. Endocrine causes of nonalcoholic fatty liver disease. World J Gastroenterol 2015, 21,11053-11076. Available online: https://www.ncbi.nlm.nih.gov/pubmed/26494962 (accessed on 11.02.2019).

7. Kumarendran, B., O'Reilly, M.W., Manolopoulos, K.N., Toulis, K.A., Gokhale, K.M., Sitch, A.J., Wijeyaratne, C.N., Coomarasamy, A., Arlt, W., Nirantharakumar, K. Polycystic ovary syndrome, androgen excess, and the risk of nonalcoholic fatty liver disease in women: A longitudinal study based on a United Kingdom primary care database. PLoS Med 2018, 15, e1002542. Available online: https://www.ncbi.nlm.nih.gov/pubmed/29590099 (accessed on 11.02.2019).

8. Asfari, M.M., Niyazi, F., Lopez, R., Dasarathy, S., McCullough, A.J. The association of nonalcoholic steatohepatitis and obstructive sleep apnea. Eur J Gastroenterol Hepatol 2017, 29, 1380-1384. Available online: https://www.ncbi.nlm.nih.gov/pubmed/28914694 (accessed on 11.02.2019).

9. Kälsch, J., Bechmann, L.P., Manka, P., Kahraman, A., Schlattjan, M., Marth, T., Rehbehn, K., Baba, H.A., Canbay, A. Non-alcoholic steatohepatitis occurs in celiac disease and is associated with cellular stress. Z Gastroenterol 2013, 51, 26-31. Available online: https://www.ncbi.nlm.nih.gov/pubmed/23315648 (accessed on 11.02.2019).

5)      Table 1 insert the references

Proper references were inserted into the table, including four new positions in the References list.

14. Musso, G., Cassader, M., Rosina, F., Gambino, R. Impact of current treatments on liver disease, glucose metabolism and cardiovascular risk in non‐alcoholic fatty liver disease (NAFLD): a systematic review and meta‐analysis of randomised trials. Diabetologia2012, 55, 885‐904. Available online: https://www.ncbi.nlm.nih.gov/pubmed/22278337 (accessed on 11.02.2019).

16. Haufe, S., Engeli, S., Kast, P., Böhnke, J., Utz, W., Haas, V., Hermsdorf, M., Mähler, A., Wiesner, S., Birkenfeld, A.L., Sell, H., Otto, C., Mehling, H., Luft, F.C., Eckel, J., Schulz-Menger, J., Boschmann, M., Jordan, J. Randomized comparison of reduced fat and reduced carbohydrate hypocaloric diets on intrahepatic fat in overweight and obese human subjects. Hepatology2011, 53, 1504‐1514. Available online: https://www.ncbi.nlm.nih.gov/pubmed/21400557 (accessed on 11.02.2019).

18. George, E.S., Forsyth, A., Itsiopoulos, C., Nicoll, A.J., Ryan, M., Sood, S., Roberts, S.K., Tierney, A.C. Practical Dietary Recommendations for the Prevention and Management of Nonalcoholic Fatty Liver Disease in Adults. Adv Nutr 2018, 9, 30-40. Available online: https://www.ncbi.nlm.nih.gov/pubmed/29438460 (accessed on 11.02.2019).

19. Green, C.J., Hodson, L. The influence of dietary fat on liver fat accumulation. Nutrients 2014, 6, 5018-5033. Available online: https://www.ncbi.nlm.nih.gov/pubmed/25389901 (accessed on 11.02.2019).

6)      I would also transfer the text from line 120 to 133 on pg. 4 where you can find the paragraph entitled “pharmacological  treatment”.   

The paragraph regarding vitamin supplementation was transferred to the section

3. Pharmacological Treatment, subsection 3.1 Vitamin supplementation. (Page 6, Lines 194-211).

7)      Please update references

Following Reviewers' suggestions, we updated references.

Reviewer 2 Report

This MS is dealing with dietary intervention and  pharmacological treatment in patients with NAFLD/NASH

I have several questions and requests:

1.     Abstract:  add “… and blood hypertension” in the definition of  MetS.

2.     Methods: Authors should specify if this is a narrative review or a systematic review, and which were the criteria of selection of the referenced studies.

3.     A better whole ri-organization of the MS in sub-paragraphs by means of soecific sub-headings might help to follow the pages containing busy periods

4.     Table is interesting as it summarizes available dietary recommendations.  However it might be improved:                                 

 a. Authors should indicate also in the Table the References quoted in the text       b. the two sections of the Table should be separated                                            

 c. Authors should  specify if the Table is Modified from other papers

5.     Microbiota and NAFLD:  Authors should

a.     add and discuss some systematic  reviews and meta-analysis on the effects of probiotics on NAFLD (e.g. . Ma YY, et al Effects of probiotics on nonalcoholic fatty liver disease: a meta-analysis. World J Gastroenterol. 2013 ; S Lavekar A Role of Probiotics in the Treatment of Nonalcoholic Fatty Liver Disease: A Meta-analysis. Euroasian J Hepatogastroenterol. 2017)

b.      introduce here the concept of gut liver axis and NAFLD (e.g. Poeta M, Pierri L, Vajro P. Gut-Liver Axis Derangement in Non-Alcoholic Fatty Liver Disease. Children (Basel). 2017)

6.   Pharmachological Treatment: Authors should add information on UDCA and Obeticholic acid (OCA) in NAFLD

In particular they might  say that NAFLD has been shown to be associated with abnormal  Farnesoid X receptor (FXR) activity which is a good target of OCA, and add that OCA has shown promising effects  on NASH in several studies (e.g. reviewed in Abenavoli L, Obeticholic Acid: A New Era in the Treatment of Nonalcoholic Fatty Liver Disease. Pharmaceuticals (Basel). 2018).

7.  The NASH drug pipelines on the horizon should at least be mentioned (e.g. elafibranor, selonsertib, and cenicriviroc CVC which have entered international phase 3 trials)

8.      A Figure/Cartoon showing  the main pathomechanisms that are targets of the proposed therapeutic strategies might be useful to readers

Minor issues:

there are some typos:  e.g. ref 9 nonalcoholoic; ref 24 Vitamine E; ….

Author Response

Reviewer #2

1) Abstract: add “… and blood hypertension” in the definition of MetS.

The suggested correction was added into the abstract.

“Such a therapeutic process is intended to support both: liver disease and obesity-related pathologies: insulin resistance, diabetes, dyslipidemia and blood hypertension.” (Page 1, lines 24-26)

2) Methods: Authors should specify if this is a narrative review or a systematic review, and which were the criteria of selection of the referenced studies.

We thank the Reviewer for this suggestion. Indeed, we should have mentioned what kind of review is presented. We decided to prepare a narrative review based of the current guidelines of the scientific societies (American Association for the Study of Liver Diseases, European Association for the Study of the Liver) enriched by the latest literature on the field regarding both non- and pharmacological treatment of NAFLD.

We introduced the proper information into the text.

“In this narrative review, based on the current guidelines supplemented with a review of the latest literature, we present possible therapeutic options that include lifestyle interventions and pharmacological treatment.” (Page 2, lines 77-79)

3) A better whole re-organization of the MS in sub-paragraphs by means of specific sub-headings might help to follow the pages containing busy periods

Following the Reviewer's suggestions, we re-organised the manuscript dividing it into sub-sections.

4)   Table is interesting as it summarizes available dietary recommendations.  However it might be improved:                                 

a. Authors should indicate also in the Table the References quoted in the text       

Proper references were inserted into the table, including four new positions in the References list.

14. Musso, G., Cassader, M., Rosina, F., Gambino, R. Impact of current treatments on liver disease, glucose metabolism and cardiovascular risk in non‐alcoholic fatty liver disease (NAFLD): a systematic review and meta‐analysis of randomised trials. Diabetologia2012, 55, 885‐904. Available online: https://www.ncbi.nlm.nih.gov/pubmed/22278337 (accessed on 11.02.2019).

16. Haufe, S., Engeli, S., Kast, P., Böhnke, J., Utz, W., Haas, V., Hermsdorf, M., Mähler, A., Wiesner, S., Birkenfeld, A.L., Sell, H., Otto, C., Mehling, H., Luft, F.C., Eckel, J., Schulz-Menger, J., Boschmann, M., Jordan, J. Randomized comparison of reduced fat and reduced carbohydrate hypocaloric diets on intrahepatic fat in overweight and obese human subjects. Hepatology2011, 53, 1504‐1514. Available online: https://www.ncbi.nlm.nih.gov/pubmed/21400557 (accessed on 11.02.2019).

18. George, E.S., Forsyth, A., Itsiopoulos, C., Nicoll, A.J., Ryan, M., Sood, S., Roberts, S.K., Tierney, A.C. Practical Dietary Recommendations for the Prevention and Management of Nonalcoholic Fatty Liver Disease in Adults. Adv Nutr 2018, 9, 30-40. Available online: https://www.ncbi.nlm.nih.gov/pubmed/29438460 (accessed on 11.02.2019).

19. Green, C.J., Hodson, L. The influence of dietary fat on liver fat accumulation. Nutrients 2014, 6, 5018-5033. Available online: https://www.ncbi.nlm.nih.gov/pubmed/25389901 (accessed on 11.02.2019).

b. the two sections of the Table should be separated   

Following the Reviewer's suggestion Table 1 was divided into two sections A & B.                                    

c. Authors should  specify if the Table is Modified from other papers

Table 1 was elaborated by the Authors based on the available American Association for the Study of Liver Diseases and European Association for the Study of the Liver guidelines as well as on the current literature, and this information was introduced into the table's caption.

Table 1. Summary of dietary recommendations in non-alcoholic fatty liver disease (NAFLD, own elaboration).

(Page 3, lines 92-93)

5)     Microbiota and NAFLD:  Authors should

a.     add and discuss some systematic  reviews and meta-analysis on the effects of probiotics on NAFLD (e.g. . Ma YY, et al Effects of probiotics on nonalcoholic fatty liver disease: a meta-analysis. World J Gastroenterol. 2013 ; S Lavekar A Role of Probiotics in the Treatment of Nonalcoholic Fatty Liver Disease: A Meta-analysis. Euroasian J Hepatogastroenterol. 2017)

Following the Reviewer's suggestion, we added [ref no 46 & 47] and discussed the papers mentioned above into the manuscript

“2.5. Probiotics and prebiotics.

In interventional studies supplementation with probiotics (Lactobacillus and Bifidobacterium) efficiently reduced fatty acids synthesis, metabolic endotoxaemia and inflammation in animal models of NAFLD [10]. In meta-analyses, probiotic therapies effectively reduced aminotransferases, total cholesterol, triglycerides and pro-inflammatory cytokines serum levels as well as improved insulin sensitivity and ultrasound liver image in NAFLD patients [11,46,47]. However, the effect of the therapy may vary depending on the bacterial strains and regimen of treatment [47]. Prebiotics have also been found to have a direct influence on lipids and carbohydrates metabolism. Beneficial effects of, e.g. oligofructose and inulin on glucose, glycated haemoglobin, triglycerides, total and LDL cholesterol level as well as on transaminase activity in patients with type 2 diabetes and NAFLD were observed [10,31]. Additionally, oligofructose supplementation promoted weight loss irrespective of patients' lifestyle [32]. Similarly, in a randomised trial, addition of synbiotics (combining probiotics and prebiotics) to the lifestyle intervention led to significant reduction hepatic steatosis and fibrosis and decrease in serum glucose, triglycerides and inflammatory mediators levels [48]”. (Page 5, lines 156-170)

b.      introduce here the concept of gut liver axis and NAFLD (e.g. Poeta M, Pierri L, Vajro P. Gut-Liver Axis Derangement in Non-Alcoholic Fatty Liver Disease. Children (Basel). 2017)

We introduced a paragraph regarding the “gut-liver axis” into the Introduction with proper reference [no 12].

“According to the “multiple hit” hypothesis, in obese individuals with excessive lipid accumulation in the liver (“first hit”), derangement of the gut-liver axis acts as a “second hit” leading to the progression of liver damage, inflammation and subsequent fibrosis [11]. Intestinal dysbiosis might contribute to NAFLD progression in several ways: (i) by increasing energy extraction from diet due to the altered capacity to digest and ferment complex polysaccharides; (ii) by damage caused to intestinal epithelium by bacterial production of ethanol; (iii) by translocation of bacteria and endotoxin to the portal circulation and activation of pro-inflammatory signaling via toll-like receptors (TLRs); (iv) by modifications of bile acids synthesis; (v) by decrease in choline metabolism resulting in reduced liver export of very low density lipoproteins [12].” (Page 2, lines 67-75).

6) & 7)   Pharmachological Treatment: Authors should add information on UDCA and Obeticholic acid (OCA) in NAFLD. In particular they might  say that NAFLD has been shown to be associated with abnormal  Farnesoid X receptor (FXR) activity which is a good target of OCA, and add that OCA has shown promising effects  on NASH in several studies (e.g. reviewed in Abenavoli L, Obeticholic Acid: A New Era in the Treatment of Nonalcoholic Fatty Liver Disease. Pharmaceuticals (Basel). 2018). The NASH drug pipelines on the horizon should at least be mentioned (e.g. elafibranor, selonsertib, and cenicriviroc CVC which have entered international phase 3 trials)

Following the Reviewer's suggestions, we added a separate section regarding novel therapeutic perspectives in NAFLD and discussed studies regarding the compounds mentioned above in sections

3.3 Compounds modifying lipid profile.

“Several novel compounds interfering with lipid metabolism are being tested for their efficacy in NAFLD treatment. Aramchol, a cholic-arachidic acid conjugate, acting as stearoyl-CoA desaturase (SCD) inhibitor was found to reduce hepatic fat accumulation both in animal and in human studies [63], and the drug is being tested in on biopsy-proven NASH patients without cirrhosis in phase 2b clinical trial [64]. GS0976, an inhibitor of the acetyl-CoA carboxylase (ACC), a key regulator of fatty acid metabolism, was also found to decrease hepatic fat content and levels of serum markers of hepatic fibrosis in NASH patients [64].

Interference with liver lipid metabolism in order to improve NAFLD course is extensively investigated in pre-clinical studies, e.g. inhibition of diacylglycerol acyltransferase 2 (DGAT), that catalyses the final step in triglyceride synthesis, with antisense oligonucleotides in rats with diet-induced NAFLD significantly reduced hepatic steatosis and improved insulin sensitivity [65]. This concept may represent a promising therapeutic perspective.” (Page 9, lines 235-246)

3.5. Novel therapeutic perspectives.

 “Recent years unrevealed novel mechanisms in the pathogenesis of NAFLD and created novel therapeutic possibilities in its treatment that include, among others, activation of farnesoid X receptor (FXR) as well as interference with apoptotic, fibrotic and inflammatory pathways.

Farnesoid X receptor agonists

                FXR is a nuclear receptor, widely expressed in liver, regulated by bile acids which abnormal activity is associated with NAFLD. Obeticholic acid (OCA) is a first-in-class selective FXR agonist with anticholestatic and hepato-protective properties, registered for the treatment of primary biliary cholangitis [67]. In phase 3 clinical trials performed in patients with NAFLD and type 2 diabetes and in individuals with NASH proven by biopsy OCA was found to enhance insulin sensitivity, control glucose homeostasis, modulate lipid metabolism, and exert anti-inflammatory and anti-fibrotic effects in the liver. The most common adverse effects of OCA treatment include gastrointestinal problems, pruritus, fatigue, headache, increase in LDL-cholesterol and a decrease in HDL-cholesterol and triglycerides. [67]. Therefore non-bile acid synthetic FXR agonists that have the potential to provide favourable metabolic effects without increasing these side effects are currently assessed in phase 2 trials [64].

                Another bile acid – ursodeoxycholic acid (UDCA), produced naturally by intestine bacteria, has also been tested for its utility in NAFLD treatment. In pre-clinical studies, UDCA was found to exert anti-apoptotic, anti-oxidant and anti-inflammatory effects. However, in clinical trials in patients with NASH, UDCA has failed to have a significant influence on liver inflammation or fibrosis, therefore currently it is not recommended by the guidelines [1,68]. Nevertheless, it has been suggested that a combination of UDCA with other agents, such as vitamin E and/or omega-3 might have an additive effect in diminishing NASH-associated fibrosis [69].

Dual PPAR agonists.

 Another new trend in NAFLD treatment represent dual PPAR agonists. Saroglitazar, a dual PPARα/γ agonist used for the treatment of dyslipidemia in diabetic patients and elafibranor, a dual agonist of PPARα/δ which was found to be effective in improvement of steatosis, inflammation, and fibrosis in mouse models of NAFLD are currently under assessment in phase 2 trials [64].

 Compounds interfering with apoptotic pathways

                Anti-apoptotic agents such as Selonsertib, an inhibitor of the apoptosis signal-regulating kinase 1 (ASK1 – a key player in the pathways leading to liver apoptosis and fibrosis) was evaluated in phase 2 trials in NASH patients with moderate-to-severe liver fibrosis (stages 2/3). It led to significant regression of fibrosis stage, reduction of liver stiffness and fat content and diminished the risk of progression to cirrhosis. Thus, international phase 3 trials evaluating selonsertib in NASH patients with stage 3 are ongoing [64].

In turn, mTORC1 inhibitor (rapamycin) was found to improve liver steatosis in high fat diet-fed mice; however, the treatment resulted in increased production of interleukin 6 and activation of signal transducer and activator of transcription 3 (STAT3) pathway that may enhance HCC development [70].

Anti-fibrotic and anti-inflammatory compounds

                Since fibrosis stage determines mortality in NASH patients, effective anti-fibrotic treatment could improve the course and prognosis of the disease. Therefore several compounds with the anti-inflammatory and anti-fibrotic potential have been considered for the treatment of advanced NASH.

                Historically, pentoxyphiline (a non-selective phosphodiesterase inhibitor) that via inhibition of tumour necrosis alpha (TNF-α) exerts an anti-inflammatory effect and in this way may decrease aminotransferases activity is controversial as well as the influence of this drug on the histological image of the liver [71,72].

                Nowadays, anti-fibrotic compound emricasan, a caspase inhibitor that improves fibrosis in murine models of NASH is being evaluated for its utility in phase 2b study in patients with NASH (stage 1-3) including those with cirrhosis and severe portal hypertension [73]. The similar anti-fibrotic effect was reported in murine models for the Vascular adhesion protein-1 (VAP-1) inhibitors. VAP-1 is responsible for transmission of pro-fibrogenic and pro-inflammatory stimuli. Thus, inhibitors targeting VAP-1 are under evaluation in clinical trials in NASH patients [74].

                Another anti-inflammatory and anti-fibrotic compound Cenicriviroc (CVC), a C–C motif chemokine receptor-2/5 (CCR2/5) antagonist that also improves insulin sensitivity via inhibition of macrophage recruitment into adipose tissue was reported to improve fibrosis without worsening NASH compared with placebo [75] and is being further tested for clinical application.

                Available and future therapies for different stages of NAFLD are summarised in Figure 1.” (Pages 9-10, lines 257-320).

8.      A Figure/Cartoon showing  the main pathomechanisms that are targets of the proposed therapeutic strategies might be useful to readers

An appropriate figure (Figure 1) was prepared and included into the corrected version of the manuscript

Minor issues:

there are some typos:  e.g. ref 9 nonalcoholoic; ref 24 Vitamine E;

The typos were corrected.

Reviewer 3 Report

Jeznach-Steinhagen et al. review on Dietary and pharmacological treatments of NAFLD is timely and informative to the readers of NALFD field. However, author should not limit the literature review of therapeutic option to the diabetic patients (table 2). There are pharmacological drug (such as Oltipraz, Saroglitazar, etc.) under clinical trials in NAFLD/NASH patients. Hence, authors should broaden their focus on NAFLD/NASH patients and cover more therapeutic options targeting various mechanism such as natural antioxidants, DGAT inhibitors, mTOR inhibitors etc.

Author Response

Reviewer #3

Jeznach-Steinhagen et al. review on Dietary and pharmacological treatments of NAFLD is timely and informative to the readers of NALFD field. However, author should not limit the literature review of therapeutic option to the diabetic patients (table 2). There are pharmacological drug (such as Oltipraz, Saroglitazar, etc.) under clinical trials in NAFLD/NASH patients. Hence, authors should broaden their focus on NAFLD/NASH patients and cover more therapeutic options targeting various mechanism such as natural antioxidants, DGAT inhibitors, mTOR inhibitors etc.

Following the Reviewer's suggestions we added information regarding natural antioxidants:

“Cinnamon and turmeric are also believed to improve insulin sensitivity, decrease fasting glucose levels by reduction of hepatic gluconeogenesis, lower HOMA-IR, improve lipid profile and decrease transaminase activity [28,29, 45].” (Page 5, lines 152-154)

“Among other substances with anti-oxidant properties glutathione was found to decrease ALT levels and hepatic steatosis in NAFLD patients, however, large-scale clinical trials are needed to verify its finding [52].” (Page 6, lines 209-211)

- compounds modifying lipid profile:

3.3 Compounds modifying lipid profile.

“Several novel compounds interfering with lipid metabolism are being tested for their efficacy in NAFLD treatment. Aramchol, a cholic-arachidic acid conjugate, acting as stearoyl-CoA desaturase (SCD) inhibitor was found to reduce hepatic fat accumulation both in animal and in human studies [63], and the drug is being tested in on biopsy-proven NASH patients without cirrhosis in phase 2b clinical trial [64]. GS0976, an inhibitor of the acetyl-CoA carboxylase (ACC), a key regulator of fatty acid metabolism, was also found to decrease hepatic fat content and levels of serum markers of hepatic fibrosis in NASH patients [64].

Interference with liver lipid metabolism in order to improve NAFLD course is extensively investigated in pre-clinical studies, e.g. inhibition of diacylglycerol acyltransferase 2 (DGAT), that catalyses the final step in triglyceride synthesis, with antisense oligonucleotides in rats with diet-induced NAFLD significantly reduced hepatic steatosis and improved insulin sensitivity [65]. This concept may represent a promising therapeutic perspective.” (Page 9, lines 235-246)

- and a separate section regarding novel therapeutic perspectives in NAFLD.

3.5. Novel therapeutic perspectives.

“Recent years unrevealed novel mechanisms in the pathogenesis of NAFLD and created novel therapeutic possibilities in its treatment that include, among others, activation of farnesoid X receptor (FXR) as well as interference with apoptotic, fibrotic and inflammatory pathways.

Farnesoid X receptor agonists

                FXR is a nuclear receptor, widely expressed in liver, regulated by bile acids which abnormal activity is associated with NAFLD. Obeticholic acid (OCA) is a first-in-class selective FXR agonist with anticholestatic and hepato-protective properties, registered for the treatment of primary biliary cholangitis [67]. In phase 3 clinical trials performed in patients with NAFLD and type 2 diabetes and in individuals with NASH proven by biopsy OCA was found to enhance insulin sensitivity, control glucose homeostasis, modulate lipid metabolism, and exert anti-inflammatory and anti-fibrotic effects in the liver. The most common adverse effects of OCA treatment include gastrointestinal problems, pruritus, fatigue, headache, increase in LDL-cholesterol and a decrease in HDL-cholesterol and triglycerides. [67]. Therefore non-bile acid synthetic FXR agonists that have the potential to provide favourable metabolic effects without increasing these side effects are currently assessed in phase 2 trials [64].

                Another bile acid – ursodeoxycholic acid (UDCA), produced naturally by intestine bacteria, has also been tested for its utility in NAFLD treatment. In pre-clinical studies, UDCA was found to exert anti-apoptotic, anti-oxidant and anti-inflammatory effects. However, in clinical trials in patients with NASH, UDCA has failed to have a significant influence on liver inflammation or fibrosis, therefore currently it is not recommended by the guidelines [1,68]. Nevertheless, it has been suggested that a combination of UDCA with other agents, such as vitamin E and/or omega-3 might have an additive effect in diminishing NASH-associated fibrosis [69].

Dual PPAR agonists.

 Another new trend in NAFLD treatment represent dual PPAR agonists. Saroglitazar, a dual PPARα/γ agonist used for the treatment of dyslipidemia in diabetic patients and elafibranor, a dual agonist of PPARα/δ which was found to be effective in improvement of steatosis, inflammation, and fibrosis in mouse models of NAFLD are currently under assessment in phase 2 trials [64].

 Compounds interfering with apoptotic pathways

                Anti-apoptotic agents such as Selonsertib, an inhibitor of the apoptosis signal-regulating kinase 1 (ASK1 – a key player in the pathways leading to liver apoptosis and fibrosis) was evaluated in phase 2 trials in NASH patients with moderate-to-severe liver fibrosis (stages 2/3). It led to significant regression of fibrosis stage, reduction of liver stiffness and fat content and diminished the risk of progression to cirrhosis. Thus, international phase 3 trials evaluating selonsertib in NASH patients with stage 3 are ongoing [64].

In turn, mTORC1 inhibitor (rapamycin) was found to improve liver steatosis in high fat diet-fed mice; however, the treatment resulted in increased production of interleukin 6 and activation of signal transducer and activator of transcription 3 (STAT3) pathway that may enhance HCC development [70].

Anti-fibrotic and anti-inflammatory compounds

                Since fibrosis stage determines mortality in NASH patients, effective anti-fibrotic treatment could improve the course and prognosis of the disease. Therefore several compounds with the anti-inflammatory and anti-fibrotic potential have been considered for the treatment of advanced NASH.

                Historically, pentoxyphiline (a non-selective phosphodiesterase inhibitor) that via inhibition of tumour necrosis alpha (TNF-α) exerts an anti-inflammatory effect and in this way may decrease aminotransferases activity is controversial as well as the influence of this drug on the histological image of the liver [71,72].

                Nowadays, anti-fibrotic compound emricasan, a caspase inhibitor that improves fibrosis in murine models of NASH is being evaluated for its utility in phase 2b study in patients with NASH (stage 1-3) including those with cirrhosis and severe portal hypertension [73]. The similar anti-fibrotic effect was reported in murine models for the Vascular adhesion protein-1 (VAP-1) inhibitors. VAP-1 is responsible for transmission of pro-fibrogenic and pro-inflammatory stimuli. Thus, inhibitors targeting VAP-1 are under evaluation in clinical trials in NASH patients [74].

                Another anti-inflammatory and anti-fibrotic compound Cenicriviroc (CVC), a C–C motif chemokine receptor-2/5 (CCR2/5) antagonist that also improves insulin sensitivity via inhibition of macrophage recruitment into adipose tissue was reported to improve fibrosis without worsening NASH compared with placebo [75] and is being further tested for clinical application.

                Available and future therapies for different stages of NAFLD are summarised in Figure 1.” (Pages 9-10, lines 257-320)

Round  2

Reviewer 2 Report

The Authors reviewed the MS according to this Referee's suggestions.

Minor/Typos

Abstract: line 24 “to support both: liver disease …”   à   “to support both liver disease …”

Introduction : line 48-50 à “ ….. with differences according to the diagnostic method…… “, add BMI as well

Line 64 “…. and celiac disease may promote the progress of….” Should be “…. and celiac disease may associate with or promote the progress of….”

Table 1: Summary of dietary recommendations in non-alcoholic fatty liver disease (NAFLD, own  elaboration) should be “Summary of dietary recommendations in non-alcoholic fatty liver disease (NAFLD)."

Table 1: Authors please separate graphically better the groups

Line 259: “ .... years unrevealed ….. “ should be à ".. years revealed, unveiled, uncovered….".

Legend to Figure 1; Authors please add the abbreviation of HCC

Author Response

Reviewer #2

Following Reviewer’s suggestions we corrected typos and introduced minor revisions into the text.

Minor/Typos

1)    Abstract: line 24 “to support both: liver disease …”   à   “to support both liver disease …”

“Such a therapeutic process is intended to support both liver disease and obesity-related pathologies: insulin resistance, diabetes, dyslipidemia and blood hypertension.” (Page 1, lines 24-25)

2)    Introduction : line 48-50 à “ ….. with differences according to the diagnostic method…… “, add BMI as well

“It is diagnosed in 15-40%, of the adult population (with differences according to the diagnostic method, age, sex ethnicity, and body mass index (BMI), twice more often in men than in women) [2].” (Page 2, lines 48-50)

3)    Line 64 “…. and celiac disease may promote the progress of….” Should be “…. and celiac disease may associate with or promote the progress of….”

“In addition, co-existence of other medical conditions such as hypothyroidism, hypogonadism, obstructive sleep apnea, polycystic ovary syndrome and celiac disease may associate with or promote the progress of NAFL into NASH [5-9].” (Page 2, lines 62-64)

4)    Table 1: Summary of dietary recommendations in non-alcoholic fatty liver disease (NAFLD, own  elaboration) should be “Summary of dietary recommendations in non-alcoholic fatty liver disease (NAFLD)."

Table 1. Summary of dietary recommendations in non-alcoholic fatty liver disease (NAFLD). (Page 3, lines 91)

5)    Table 1: Authors please separate graphically better the groups

We have made efforts to improve the graphic appearance of Table 1. (Pages 3-4)

6)    Line 259: “ .... years unrevealed ….. “ should be à ".. years revealed, unveiled, uncovered….".

“Recent years revealed novel mechanisms in the pathogenesis of NAFLD and created novel therapeutic possibilities in its treatment that include, among others, activation of farnesoid X receptor (FXR) as well as interference with apoptotic, fibrotic and inflammatory pathways.” (Page 9, lines 256-258)

7)    Legend to Figure 1; Authors please add the abbreviation of HCC

“Figure 1. Available and future therapies for different stages of non-alcoholic fatty liver disease (modified based on Sumida & Yoneda, 2018 [64])

ACCi – acetyl-CoA carboxylase inhibitors, DPP-IVi – dipeptidyl peptidase-4 inhibitors, FXR – farnesoid X receptor, GLP-1 – glucagon-like peptide-1, HCC – hepatocellular carcinoma, PPARs – peroxisome proliferator-activated receptors, SGLT2i – sodium glucose co-transporter 2 inhibitors, VAP-1i – vascular adhesion protein-1 inhibitors” (Page 11, lines 318-323)

Reviewer 3 Report

Authors have modified and improved the manuscript. 

Author Response

Reviewer 3#

"Authors have modified and improved the manuscript."

We thank the Reviewer for the acceptance of the present version of the manuscript.